# Herlyn-Werner-Wunderlich Syndrome: Case Report and Review of the Literature

**DOI:** 10.3390/diagnostics12102466

**Published:** 2022-10-12

**Authors:** Anca Maria Panaitescu, Gheorghe Peltecu, Nicolae Gică

**Affiliations:** 1Department of Obstetrics and Gynecology, Carol Davila University of Medicine and Pharmacy, 020021 Bucharest, Romania; 2Filantropia Clinical Hospital, 011132 Bucharest, Romania

**Keywords:** Herlyn-Werner-Wunderlich syndrome, diagnosis, treatment, prognosis

## Abstract

Herlyn-Werner-Wunderlich (HWW) syndrome is a very rare congenital malformation of the urogenital tract involving both the müllerian and the wolffian ducts, and it consists of the association of didelphys uterus, ipsilateral obstructed hemivagina, and ipsilateral kidney agenesis. Its etiology is related to the abnormal development of two embryonic structures—mesonephros and paramesonephros—although its precise mechanism is not known, but theories involving signaling molecules and gene expression are studied for potential explanations. Because of its rarity, there is limited literature on this subject. We present a case diagnosed in our department and elaborate on management. In HWW syndrome, symptoms appear after menarche and include pelvic pain, with progressive intensity due to the developing of hematocolpos. Menstruation may be present or absent depending on the type of anomaly. The use of magnetic resonance imaging (MRI) is the most recommended method of investigation since, in most cases, at this age, sexual life has not started yet and transvaginal ultrasound cannot be used. Surgical treatment in our case consisted of an exploratory laparoscopy followed by a vaginal surgical approach to resect the vaginal septum of the obstructed hemivagina.

## 1. Introduction

Some authors consider that the first and incomplete description of what is called today Herlyn-Werner-Wunderlich syndrome was published in 1922 [1]. The triad was fully communicated by Herlyn and Werner in 1971, who presented a case with double uterus, ipsilateral blind vagina, and renal agenesis [2]. Wunderlich, in 1976, published a case with bicornuate uterus and ipsilateral blind hemivagina, associated with an isolated hematocervix and right kidney aplasia [3].

The development of the female reproductive system is a very complex process. Understanding the embryology of the female reproductive tract will help clinicians diagnose the congenital anomalies of these organs. In the first 6 weeks of the embryo’s life, the genital system is considered undifferentiated. Sexual differentiation of the reproductive ducts in female starts early in the embryo’s life and is based on two pairs of genital ducts, situated on either side of the midline. These are the mullerian (paramesonephros) and the wolffian (mesonephros) ducts [4]. Sexual differentiation will evolve under the influence of gonadal hormones, but genetic information will control the differentiation of the embryonic structures. An essential hormone regulating sexual differentiation is the anti-mullerian hormone (AMH). It is produced by Sertoli cells and will stop the development of female genital organs. In the absence of AMH, the mullerian ducts will persist and continue to grow caudally, generating the uterus and the round ligaments [4]. The caudal parts of the paramesonephros will fuse at 10 weeks’ gestation, before reaching the urogenital sinus and forming the uterovaginal canal, which will further connect to the urogenital sinus. The uterovaginal canal is a tube with one lumen and a septum in its upper part. By 10 weeks of gestation, septum reabsorption begins, and it will be completed by 20 weeks. The upper part of the mullerian ducts that did not fuse will become the fallopian tubes, and the lowest part will generate the upper 1/3 of the vagina. The lower part of the vagina will develop from the urogenital sinus. The development and differentiation of the paramesonephros is regulated by different signaling molecules and gene expression (examples are EMX2, HOXA13, MIM1, PAX2, and Wnt). These molecules are essential in mullerian and wolffian duct epithelium formation. If these transcription factors are absent, the agenesis of the Mullerian ducts, the absence of the kidneys, and the agenesis of the reproductive tract occurs. Moreover, recently, patients with Lim1 mutation have presented with abnormal fallopian tubes, uterine aplasia, and infertility. Any factor that interferes with this complex process could generate variate types of congenital anomalies [4]. The influence of wolffian ducts significantly interferes with the later stage of development of the mullerian ducts, especially their correct elongation. Mullerian anomalies may be the consequence of three processes, acting alone or combined: arrest of the development of the ducts, their failure of fusion, or the failure of the reabsorption of the medial septum [5]. Didelphy uterus is classified as class III of mullerian ducts anomalies, according to the American Society of Reproductive Medicine [6].

## 2. Case Report

We present a case of HWWS diagnosed in our department in a 16-years-old girl who presented for severe left-lower quadrant pain, nausea, and urinary symptoms. Her menarche started the year before presentation, and monthly bleeding was regular; lasting 4 days; and was accompanied by increasing abdominal pain, nausea, urinary symptoms, and sometimes vomiting. She was not sexually active. During her first gynecology referral from another institution, she was recommended contraceptive pills for severe dysmenorrhea, which she refused. In the first visit in our hospital, the physical examination of the abdomen revealed mild abdominal tenderness of the left lower abdominal quadrant. Abdominal ultrasound (US) was performed but was considered not sufficiently informative; we only observed a voluminous cystic mass corresponding to hematocolpos. Therefore, after a careful explanation of the procedure and counseling, the girl and her parents decided to accept trans-rectal US examination. This revealed a didelphys uterus with a mild hematometra of the left hemiuterus, normal right hemiuterus and an impressive left-side collection, inferior and in connection with the left hemiuterus, corresponding to a hematocolpos. Both ovaries were present and of normal aspect and size (Figure 1). To better describe the findings and in view of the frequent association of congenital anomalies of the reproductive system with renal anomalies, an MRI was recommended and performed. The pelvic MRI showed two clearly separated hemiuteri (didelphys uterus), with mild left hematometra and normal right hemiuterus, mild left hematosalpinx, and an impressive left hemivagina, connected to the left hemiuterus, distended by a content whose signal was similar to methemoglobin, suggesting an obstructed hematocolpos (Figure 2A,B). Blood analysis was within normal range, except CA 125, which measured 89, 62 u/mL (more than the normal range for that age).

The MRI confirmed the US findings and described the absence of the left kidney. Thus, the diagnosis of Herlyn-Werner-Wunderlich syndrome was clearly established. The patient and her parents were carefully counselled, and they accepted a surgical treatment recommendation, consisting of an exploratory laparoscopy followed by a vaginal surgical approach to resect the vaginal septum of the obstructed hemivagina. Laparoscopy revealed a modified bloody discharge suggestive of endometriosis, covering partially the peritoneum of the uterus, tubes, bladder, and the pouch of Douglas, which was closed by filmy adhesions. Posterior to the left hemiuterus and extending to the midline, the upper pole of the left hemotocolpos was seen. The left hemiuterus was slightly distended, and the left tube moderately so. Both ovaries were endometrioma-free, and the right tube was normal.

Before starting the vaginal approach, a small hymenotomy was performed to have access into the vagina. The left hemivaginal wall bulged to the midline (a schematic representation of the anomaly and its treatment is presented in Figure 3). Two traction sutures were passed through the vaginal mucosa, above and under the greatest bulging, at a 5 cm distance, and the vaginal wall was incised, followed by the evacuation of 350 mL of a “chocolate-like” content (Figure 4). An adequate excision of the vaginal wall was performed. An 18 gauge Foley catheter inflated with saline was left in the cavity for 24 h in order to facilitate drainage.

The surgery was uneventful, and the patient was discharged next day. At 20 months after surgery, she is asymptomatic, and vaginal US did not show any sign of stenosis of the excision site. She is sexually active.

## 3. Discussion

Herein, we describe a case of Herlyn-Werner-Wunderlich syndrome and its management. As previously reported in the literature, with HWWS, symptoms start with or early after menarche. The most common symptom is abdominal pelvic pain with each menstruation, lasting 3 to 5 days, associated with nausea and vomiting. Usually HWWS is diagnosed because of dysmenorrhea requiring anti-inflammatory drugs. In rare cases, an abdominal mass can be palpated in one of the lower abdominal quadrants and can be visualized on abdominal US [7]. This happens in delayed cases when young girls menstruate regularly and is a sign of complications related to the obstructed hemivagina. Establishing an earlier diagnosis will prevent complications such as hematometra, hematosalpinx, and endometriosis [8].

When a renal anomaly is detected in the first years of life, it would be useful to recommend a pelvic MRI to detect possible uterine malformations before menarche. This will allow possible surgical treatment before complications associated with menstruation develop. MRI is the most sensitive imaging investigation for the soft tissue of the pelvis. It could identify didelphy uterus, hematocolpos, hematometra, hematosalpinx, and ovarian or pelvic endometriosis and is especially indicated when vaginal US could not be performed. MRI will also identify the renal abnormalities.

Abdominal US can reveal the absence of one kidney and the compensatory hypertrophy of the opposite one. Vaginal US is very useful for the diagnosis of HWWS when it can be performed. When MRI is not available and vaginal US can not be performed, transrectal US could offer the same valuable information. It is important to be preceded by careful counseling and informed consent. The type of vaginal malformation, obstructed or non-obstructed, is important for diagnosis. The age of diagnosis, in complete hemivaginal obstruction is 12.8 years compared to 20.6 years in cases of incomplete obstruction [9,10].

HWWS is classified in two classes: class 1, with a completely obstructed hemivagina, and class 2, including patients with an incompletely obstructed hemivagina, according to Zhu L et al. [11].

The genital and the urinary systems have a common origin, represented by the mesodermal ridge. The development of the urinary system is based on the normal development of the mesonephrotic system. An anomaly of the differentiation of the wolffian and mullerian ducts may be significantly associated with kidney anomalies. The most common anomaly associated with didelph uterus is kidney agenesis, found in 30% of cases [12,13,14]. Kidney anomalies are associated with ipsilateral obstructive mullerian anomalies in 50% of cases [14]. This association is more frequent on the right side of the body [15,16]. This association between renal and genital anomalies should be kept in mind, and when a type of malformation is detected, the other system should be investigated in order to identify a potential malformation. However, screening for HWWS is not encouraged at puberty, given the low incidence of this anomaly.

The standard treatment of the obstructed hemivagina is the surgical excision of the vaginal septum. The excision should be wide to avoid stenosis and the subsequent recurrence of hematocolpos and symptoms. Surgery should be planned before menses to better identify the most prominent place for the incision of the vaginal wall. Incision and excision should be carefully performed in order to avoid the section of a thick vaginal wall and subsequent bleeding. The hemostasis of the excision margins should be interrupted and not continuous, to avoid stenosis.

When sexual activity has not started, it is recommended to try to preserve the hymen. Vaginal septotomy through a hysteroscopic approach has been reported, when the team, usually an experienced gynecologist or a pediatric surgeon, have experience with this [17,18,19]. Frequently this is not feasible; thus, hymenotomy is required. A Foley catheter of 14–16 gauge could be inserted and kept inflated in the hemivagina, above the surgical excision, to avoid stenosis and to facilitate drainage.

In cases of stenosis and the recurrence of symptoms, re-excision is recommended. In the rare cases of repeated stenosis and the recurrence of hematocolpos, hematometra, and infections, in selected cases, a hemi-hysterectomy of the ipsilateral hemiuterus could be performed [17]. The same recommendation is for patients with classification 1.2. of HWWS with cervical atresia, as the resection of the vaginal septum will not relieve symptoms related to re-stenosis and obstruction [11]. In selected cases with symptomatic patients, laparoscopy is recommended both as diagnostic and therapeutic tool. The possible complications of HWWS are hematocolpos, hematometra, hematosalpinx, pelvic endometriosis, and ovarian endometrioma. In cases with a non-obstructed vagina, a possible complication is infection (pyohematocolpos, pyometra).

Surgical treatment, when timely and successfully performed, considerably improves the quality of life.

Fertility is well-preserved in women with HWWS. In the largest study in the literature (79 cases), there were 52 pregnancies reported among 28 of the 33 women who were interested in getting pregnant (85%). The chance for pregnancy is lower in the ipsilateral hemiuterus, on the same part with the obstructed hemivagina (19 cases or 37%) and higher in the opposite hemiuterus (33 cases or 64%) [11]. One pregnancy in each of the two hemiuteri was reported in eight cases in the same study [11]. The risk of miscarriage is estimated to be 74%, of premature delivery 22%, and of cesarean section 80% [20,21]. Another study showed that the incidence of miscarriage was 23% and of premature delivery 15%, and in 62% of cases, pregnancy progressed to term and ended by uncomplicated deliveries [22]. Psychological support is very often necessary in order to avoid anxiety and emotional disorders related to the reproductive outcome [23].

## 4. Conclusions

Herlyn-Werner-Wunderlich syndrome is a very rare association of congenital anomalies. Its main symptom is increasing abdominal and pelvic pain, preceding and accompanying menstruation. Absence of menarche at adolescence will speed up the diagnosis, while the presence of menstruation could delay it.

For an accurate diagnosis, MRI is the most sensitive imaging investigation for the soft tissue of the pelvis and is especially indicated when vaginal US can not be performed. If MRI is not available, a very good option could be transrectal US. After the careful counseling of the parents and pediatric patient, transrectal US can be used as a very useful, cheap, and rapid tool for diagnosis in the hands of gynecologist. 

The standard treatment of obstructed hemivagina is the surgical resection of the vaginal septum. In cases of repeated infections related to restenosis or in cases of classification 1.2, the hemi-hysterectomy of the ipsilateral uterus could be recommended. 

Early and successful surgical treatment will clearly improve quality of life and will prevent endometriosis.

Fertility is preserved in women with HWWS. There are conflicting results regarding the incidence of miscarriage and the rate of preterm delivery, which is 15–22%. The rate of cesarean section is very high (80%). These patients required psychological support to avoid anxiety and emotional disorders related to sexual quality of life and reproductive outcome.

## Figures and Tables

**Figure 1 diagnostics-12-02466-f001:**
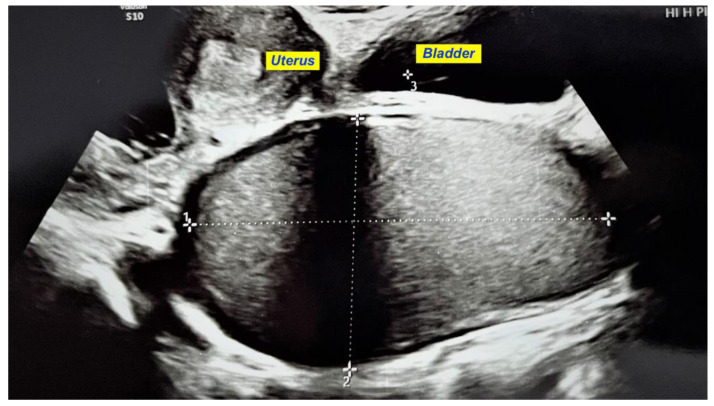
Transrectal US showing an impressive left hematocolpos and didelphys uterus with a mild hematometra of the left hemiuterus.

**Figure 2 diagnostics-12-02466-f002:**
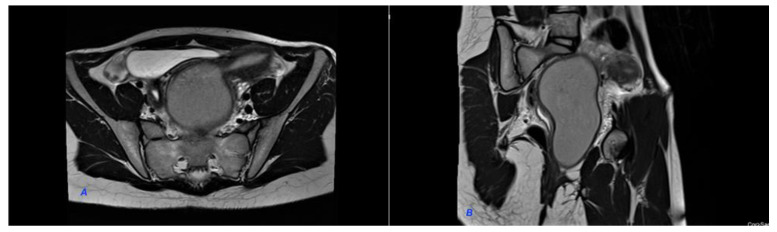
MRI of the pelvis. (**A**). Hematocolpos and left hemiuterus. (**B**). Hematocolpos and left hemiuterus. The pelvic MRI showed two clearly separated hemiuteri (didelphys uterus), with mild left hematometra and normal right hemiuterus, mild left hematosalpinx and an impressive left hemivagina, connected to the left hemiuterus, distended by a content whose signal was similar to methemoglobin, suggesting an obstructed hematocolpos.

**Figure 3 diagnostics-12-02466-f003:**
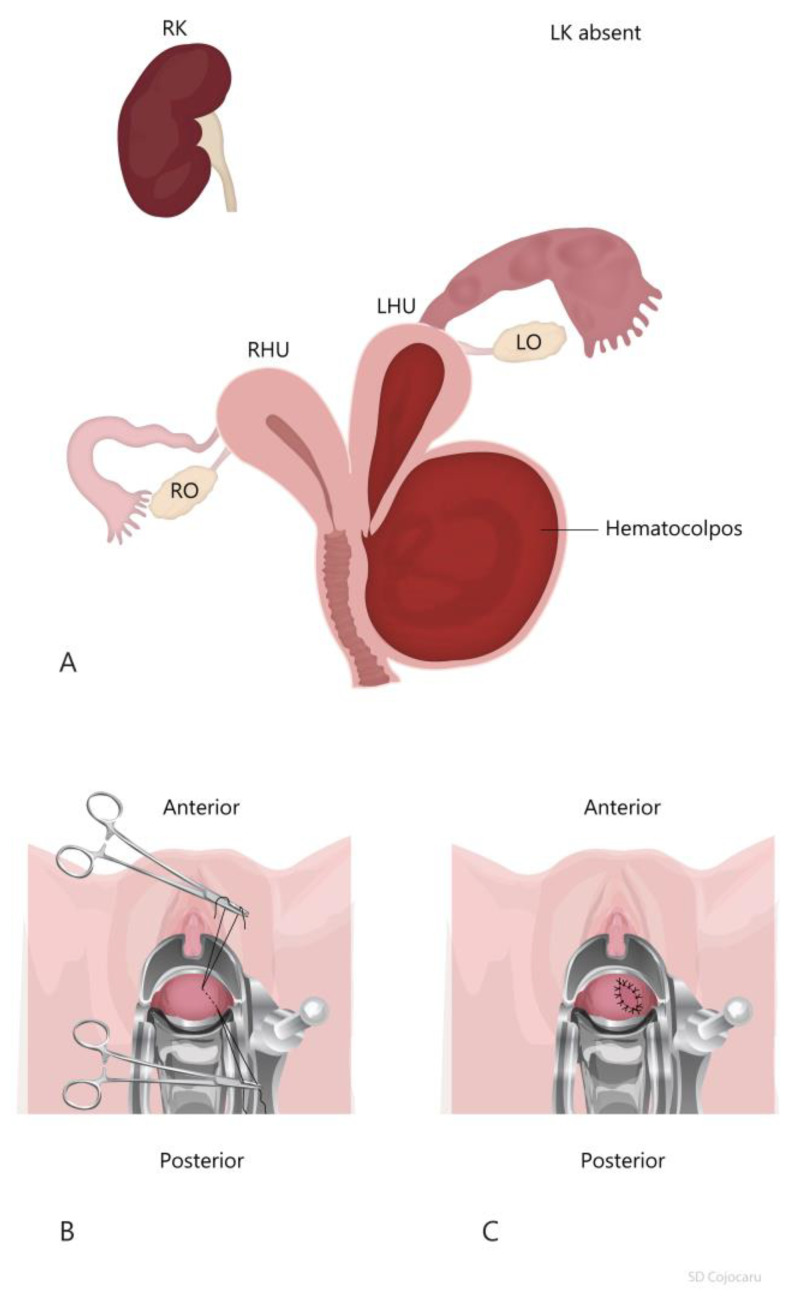
Drawing of Herlin-Werner-Wunderlich syndrome. (**A**). HWWS, classification 1.1. with obstructed left hemivagina; (**B**). traction sutures on the vaginal wall before incision; (**C**). final aspect after the partial resection of the left obstructed hemivagina.

**Figure 4 diagnostics-12-02466-f004:**
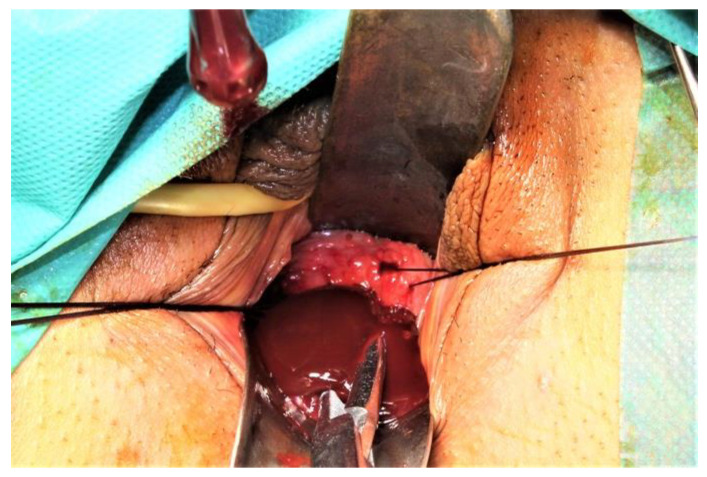
“Chocolate-like” (endometriotic fluid) content evacuated after vaginal wall incision.

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
