# Peer review of "Herlyn-Werner-Wunderlich Syndrome: Case Report and Review of the Literature"

_diagnostics, 2022, doi:10.3390/diagnostics12102466_

Round 1

Reviewer 1 Report

In this paper Panaitescu and colleagues describe an intersting case of HWW-syndrome and a review of the literature on the topic. Overall the paper misses both focus and depth. Furthermore the English grammar should be improved or proof read by a native speaker.

The introduction is interesting and relevant. However, In my opinion a more elaborate description on what is known on the genetic and molecular mechanisms of HWW (and mullerian anomalies in general) would improve the introduction. Now this is limited to one sentence (lines 53-54). A figure on the (ab)normal embryological development of the reproductive tract would be beneficial for understanding and visualising the introduction.

The description of the case is to biased and choice of words could be more objective. For example line 69: "abdominal US was not accurate". It can be accurate but that doesn't mean it's informative, or it could be that things could not be visualized. So report in full detail what was seen 1) on abdominal US, 2) transrectal ultrasound, 3) MRI 4) laparoscopy. The captions under Figure 1 and 2 should also be included in the body of the text. Figures 1 and 2 could use annotations for the described organs/anomalies.

The review of the literature should be described in more structured manner. How many papers were found and on how many patients. How did you search the literature? Use the PRISM-statement. At least one table is needed to provide a structured overview of all reported cases in literature with their anatomical variations and how they were managed.

The discussion is interesting and overall stays within the limits of scientific knowledge (or basence of it) on the topic. However some statements lack references and therefore is unclear whether they reflect personal opinion from the authors or not and on what they were based. For example: line 154-155, 167-170 (why do you need a laparoscopy?, why excise the endometrioma?, see the ESHRE guideline).

Author Response

Response to reviewer comments

In this paper Panaitescu and colleagues describe an intersting case of HWW-syndrome and a review of the literature on the topic. Overall the paper misses both focus and depth. Furthermore the English grammar should be improved or proof read by a native speaker.

Thank you for your recommendations: We performed English and punctuation edits.

The introduction is interesting and relevant. However, In my opinion a more elaborate description on what is known on the genetic and molecular mechanisms of HWW (and mullerian anomalies in general) would improve the introduction. Now this is limited to one sentence (lines 53-54). A figure on the (ab)normal embryological development of the reproductive tract would be beneficial for understanding and visualising the introduction.

Thank you for your comment. Our manuscript is interesting images type and not extensive review. For this reason the article cannot contain all the details about this type of genital malformation. We added some new information about molecular mechanisms.

“Some authors consider that a first and incomplete description of what is called today Herlyn-Werner-Wunderlich syndrome was published in 1922 [1]. The triad was fully communicated by Herlyn and Werner in 1971, who presented a case with double uterus, ipsilateral blind vagina and renal agenesis [2]. Wunderlich, in 1976, published a case with bicornuate uterus and ipsilateral blind hemivagina, associated with an isolated hematocervix and right kidney aplasia [3].

This pathology is also named OHVIRA, with the acronyms of the distinct parts of the disease, associating to didelphy uterus, obstructed hemivagina and ipsilateral renal agenesis [2].

The development of the female reproductive system is a very complex process. Knowing the embryology of the female reproductive tract will help clinicians to understand and diagnose congenital anomalies of these organs. In the first 6 weeks of the embryo’s life the genital system is considered undifferentiated. Sexual differentiation of the reproductive ducts in female starts early in the embryo’s life and is based on two pairs of genital ducts, situated on either side of the midline. These are the mullerian (paramesonephros) and the wolffian (mesonephros) ducts [4]. Sexual differentiation will evolve under the influence of gonadal hormones, but genetic information will control the differentiation of the embryonic structures. An essential hormone, regulating the sexual differentiation, is the Anti-Mullerian Hormone (AMH). It is produced by Sertoli cells and it will stop the development of female genital organs. In the absence of AMH, the mullerian ducts will persist and continue to grow caudally generating uterus and the round ligaments [4]. The caudal parts of paramesonephros will fuse at 10 weeks’ gestation, before reaching the urogenital sinus, forming the uterovaginal canal which further will connect to the urogenital sinus. The uterovaginal canal is a tube with one lumen and a septum in its upper part. By the 10 weeks’ gestation septum reasorption begins and it will be complete by 20 weeks. The upper part of the mullerian ducts which did not fuse will become fallopian tubes and the lowest part will generate the upper 1/3 of the vagina. The lower part of the vagina will develop from urogenital sinus. The development and differentiation of paramesonephros is regulated by different signaling molecules and gene expression (EMX2, HOXA13, MIM1, PAX2, Wnt). This molecules are essential in Mullerian and Wolffian duct epithelium formation. If this transcription factors are absents agenesis of the Mullerian ducts, absences of the kidneys and agenesis of reproductive tract occurs. Moreover recently the patients with Lim1 mutation presents abnormal fallopian tubes, uterine aplasia and infertility. Any factor that interferes with this complex process could generate variate types of congenital anomalies [4]. The influence of wolffian ducts significantly affects the later stage of development of the mullerian ducts, especially their correct elongation. Mullerian anomalies may be the consequence of three processes, acting alone or combined: arrest of development of the ducts, their failure of fusion or failure of reabsorption of the medial septum [5]. Didelphy uterus is classified as class III of mullerian ducts anomalies, according to the American Society of Reproductive Medicine [6].”

The description of the case is to biased and choice of words could be more objective. For example line 69: "abdominal US was not accurate". It can be accurate but that doesn't mean it's informative, or it could be that things could not be visualized. So report in full detail what was seen 1) on abdominal US, 2) transrectal ultrasound, 3) MRI 4) laparoscopy. The captions under Figure 1 and 2 should also be included in the body of the text. Figures 1 and 2 could use annotations for the described organs/anomalies.

Thank you for your recommendations: We modified the case description.

“Abdominal US was not very informative, but revealed a voluminous cystic mass corresponding to hematocolpos. After a careful explanation of the procedure and counseling, the girl and her parents decided to accept trans-rectal US examination. This revealed a didelphys uterus with a mild hematometra of the left hemiuterus, normal right hemiuterus and an impressive left side collection, inferior and in connection with left hemiuterus, corresponding to a hematocolpos. Both ovaries we present and of normal aspect and size (Figure 1). Thinking to an association with a possible renal anomaly, an MRI was recommended. The pelvic MRI showed two clearly separated hemiuteri (didelphys uterus), with mild left hematometra and normal right hemiuterus, mild left hematosalpinx and an impressive left hemivagina, connected to the left hemiuterus, distended by a content whose signal was similar to methemoglobin, suggesting an obstructed hematocolpos (Figure 2.A.B).”

The review of the literature should be described in more structured manner. How many papers were found and on how many patients. How did you search the literature? Use the PRISM-statement. At least one table is needed to provide a structured overview of all reported cases in literature with their anatomical variations and how they were managed.

Thank you for your comment: Our paper is interesting images type article (a short type of article). It is not a systematic review and for this reason we modified the title in “Herlyn-Werner-Wunderlich Syndrome. Case report and short review of the literature”

The discussion is interesting and overall stays within the limits of scientific knowledge (or basence of it) on the topic. However some statements lack references and therefore is unclear whether they reflect personal opinion from the authors or not and on what they were based. For example: line 154-155, 167-170 (why do you need a laparoscopy?, why excise the endometrioma?, see the ESHRE guideline).

Thank you for your remark: We modified those sentences. We add references.

“In selected cases with symptomatic patients, laparoscopy is recommended both as diagnostic and therapeutic tool. As a diagnostic tool laparoscopy can diagnose endometriosis when imaging is not enough accurate to detect it. When ovarian endometriomas are evident on US or MRI, laparoscopy should be performed for surgical treatment and staging. The possible complications of HWWS are hematocolpos, hematometra, hematosalpinx, pelvic endometriosis, ovarian endometrioma. In cases with non-obstructed vagina a possible complication is infection (pyohematocolpos, pyometra) [1].”

Reviewer 2 Report

Very nice , well designed and written case report

Author Response

Thank you very much for your appreciations.

Reviewer 3 Report

This is a well written and rather interesting case report on a rare congenital malformation of the female genital tract.

The paper deserves publication but a few points need to be addressed.

INTRODUCTION

Ipsilateral means "on the same side" so ipsilateral blind vagina means that the blind vagina is on the same side of the missing kidney. Therefore I would sugest the authors to write "ipsilateral blind vagina" after and not before "renal agenesia". (line 30 page1)

OHVIRA stands for Obstructed Haemi Vagina Ipsilateral Renal Agenesia, without any reference to the uterus. (line 32 page 1).

Line 38 page 1 substitute "indifferent" with "undifferentiated".

Line 52 page 2. I think that the mullerian ducts form the upper 1/3 and not the upper 4/5 of the vagina. Please see also reference 4.

CASE REPORT

Line 62 page 2. While I think this case report is indeed interesting, I would change the title into "Case report".

Line 67-68. What happened after the gynecologist prescribed a contraceptive pill? Did the patient take the pill? How did she came to your observation?  In figure 1 the endometrium does not look like she is taking a pill.

Figure 1. If this is a transrectal exam and if the left hematocolpos is posterior to the uterus (as was observed during laparoscopy), how comes that the bladder (vezica) is proximal to the probe? A drawing of the scan could be useful.

Line 107 page 4. I think a section named DISCUSSION should start here.

Finally, although the English language is perfectly clear I would suggest to have it revised by a native English speaker.

Author Response

This is a well written and rather interesting case report on a rare congenital malformation of the female genital tract.

The paper deserves publication but a few points need to be addressed.

Thank you for your appreciations.

INTRODUCTION

Ipsilateral means "on the same side" so ipsilateral blind vagina means that the blind vagina is on the same side of the missing kidney. Therefore I would sugest the authors to write "ipsilateral blind vagina" after and not before "renal agenesia". (line 30 page1)

Thank you for your recommendations: We performed the suggested remark.

OHVIRA stands for Obstructed Haemi Vagina Ipsilateral Renal Agenesia, without any reference to the uterus. (line 32 page 1).

Thank you for your recommendations: We modified accordingly.

Line 38 page 1 substitute "indifferent" with "undifferentiated".

Thank you for your recommendations: We performed the substitution.

Line 52 page 2. I think that the mullerian ducts form the upper 1/3 and not the upper 4/5 of the vagina. Please see also reference 4.

Thank you for your recommendations: We modified accordingly.

CASE REPORT

Line 62 page 2. While I think this case report is indeed interesting, I would change the title into "Case report".

Thank you for your recommendations: We modified accordingly.

Line 67-68. What happened after the gynecologist prescribed a contraceptive pill? Did the patient take the pill? How did she came to your observation?  In figure 1 the endometrium does not look like she is taking a pill.

Thank you for your comment: We added the information in the manuscript. „The family doctor referred her to the gynecologist who recommended her contraceptive pills for severe dysmenorrhea, but the treatment was refused by the patient”

Figure 1. If this is a transrectal exam and if the left hematocolpos is posterior to the uterus (as was observed during laparoscopy), how comes that the bladder (vezica) is proximal to the probe? A drawing of the scan could be useful.

Thank you for your comment: The young patient was a virgin and the probe was in the rectum. The bladder was full and above the hemiuterus. So the distended vagina (hematocolpos) is in contact with the bladder.

Line 107 page 4. I think a section named DISCUSSION should start here.

Thank you for your recommendations: We modified accordingly.

Finally, although the English language is perfectly clear I would suggest to have it revised by a native English speaker.

Thank you for your recommendations: We performed English and punctuation edits.

Reviewer 4 Report

Congratulations on your work!

Although this pathology is rare, I think the case report should be filled with more information on the background and phisiopathology of the disease and I think the conclusions paragraph could be extended more.

In figure 1 the descriptions are not in English and should be corrected. Minor English spelling errors throughout.

Author Response

Congratulations on your work!

Although this pathology is rare, I think the case report should be filled with more information on the background and phisiopathology of the disease and I think the conclusions paragraph could be extended more.

Thank you for your comment. Our manuscript is interesting images type and not extensive review. For this reason the article cannot contain all the details about this type of genital malformation. We added some new information about molecular mechanisms. The conclusion was extended.

Some authors consider that a first and incomplete description of what is called today Herlyn-Werner-Wunderlich syndrome was published in 1922 [1]. The triad was fully communicated by Herlyn and Werner in 1971, who presented a case with double uterus, ipsilateral blind vagina and renal agenesis [2]. Wunderlich, in 1976, published a case with bicornuate uterus and ipsilateral blind hemivagina, associated with an isolated hematocervix and right kidney aplasia [3].

This pathology is also named OHVIRA, with the acronyms of the distinct parts of the disease, associating to didelphy uterus, obstructed hemivagina and ipsilateral renal agenesis [2].

The development of the female reproductive system is a very complex process. Knowing the embryology of the female reproductive tract will help clinicians to understand and diagnose congenital anomalies of these organs. In the first 6 weeks of the embryo’s life the genital system is considered undifferentiated. Sexual differentiation of the reproductive ducts in female starts early in the embryo’s life and is based on two pairs of genital ducts, situated on either side of the midline. These are the mullerian (paramesonephros) and the wolffian (mesonephros) ducts [4]. Sexual differentiation will evolve under the influence of gonadal hormones, but genetic information will control the differentiation of the embryonic structures. An essential hormone, regulating the sexual differentiation, is the Anti-Mullerian Hormone (AMH). It is produced by Sertoli cells and it will stop the development of female genital organs. In the absence of AMH, the mullerian ducts will persist and continue to grow caudally generating uterus and the round ligaments [4]. The caudal parts of paramesonephros will fuse at 10 weeks’ gestation, before reaching the urogenital sinus, forming the uterovaginal canal which further will connect to the urogenital sinus. The uterovaginal canal is a tube with one lumen and a septum in its upper part. By the 10 weeks’ gestation septum reasorption begins and it will be complete by 20 weeks. The upper part of the mullerian ducts which did not fuse will become fallopian tubes and the lowest part will generate the upper 1/3 of the vagina. The lower part of the vagina will develop from urogenital sinus. The development and differentiation of paramesonephros is regulated by different signaling molecules and gene expression (EMX2, HOXA13, MIM1, PAX2, Wnt). This molecules are essential in Mullerian and Wolffian duct epithelium formation. If this transcription factors are absents agenesis of the Mullerian ducts, absences of the kidneys and agenesis of reproductive tract occurs. Moreover recently the patients with Lim1 mutation presents abnormal fallopian tubes, uterine aplasia and infertility. Any factor that interferes with this complex process could generate variate types of congenital anomalies [4]. The influence of wolffian ducts significantly affects the later stage of development of the mullerian ducts, especially their correct elongation. Mullerian anomalies may be the consequence of three processes, acting alone or combined: arrest of development of the ducts, their failure of fusion or failure of reabsorption of the medial septum [5]. Didelphy uterus is classified as class III of mullerian ducts anomalies, according to the American Society of Reproductive Medicine [6].

Herlyn-Werner-Wunderlich syndrome is a very rare association of congenital anomalies. Its main symptom is increasing abdominal and pelvic pain, preceding, and accompanying menstruation. Absence of menarche at an adolescent will speed up the diagnosis, while the presence of menstruation could delay it, as the young girl would receive contraceptive pills or anti-inflammatory drugs for a supposed dysmenorrhea.

For an accurate diagnosis, MRI is the most sensitive imaging investigation for the soft tissue of the pelvis and is especially indicated when vaginal US could not be performed. If MRI is not available, a very good option could be transrectal US. After a careful counseling of parents and pediatric patient, transrectal US can be proven as a very useful, cheap, and rapid tool for diagnosis in the hands of gynecologist.

The standard treatment of obstructed hemivagina is the surgical resection of the vaginal septum. In cases of repeated infections related to restenosis or in cases of classification 1.2, hemi-hysterectomy of the ipsilateral uterus could be recommended.

Early and successful surgical treatment will clearly improve the quality of life and will prevent endometriosis.

Fertility is preserved in women with HWWS. There are conflicting results regarding the incidence of miscarriage and the rate of preterm delivery which varies between 15-22%. The rate of cesarean section is very high (80%). This patients required psychological support to avoid anxiety and emotional disorders related to the sexual quality of life and reproductive outcome.”

In figure 1 the descriptions are not in English and should be corrected. Minor English spelling errors throughout.

Thank you for your remark: We modified the figure and the text is now in English.